# FLUKA Simulations of *Kβ*/*Kα* Intensity Ratios of Copper in Ag–Cu Alloys

**DOI:** 10.3390/ma14164462

**Published:** 2021-08-09

**Authors:** Aneta Maria Gójska, Karol Kozioł, Adam Wasilewski, Ewelina Agnieszka Miśta-Jakubowska, Piotr Mazerewicz, Jakub Szymanowski

**Affiliations:** National Centre for Nuclear Research, ul. A. Soltana 7, 05-480 Otwock, Poland; karol.koziol@ncbj.gov.pl (K.K.); adam.wasilewski@pwr.edu.pl (A.W.); ewelina.mista@ncbj.gov.pl (E.A.M.-J.); piotr.mazerewicz@ncbj.gov.pl (P.M.); jakub.szymanowski@ncbj.gov.pl (J.S.)

**Keywords:** FLUKA simulation, Kβ/Kα intensity ratios, Ag-Cu alloy

## Abstract

The numerical simulations of Cu Kα and Cu Kβ fluorescence lines induced by Rh X-ray tube and by monoenergetic radiation have been presented. The copper Kβ/Kα intensity ratios for pure elements as well as for Ag–Cu alloys have been modeled. The results obtained by use of the FLUKA code, based on the Monte-Carlo approach, have been compared to available experimental and theoretical values. A visible relationship was found between the simulated Kβ/Kα intensity ratios and the copper content of the Ag–Cu alloy: As the Cu content increases, the Kβ/Kα coefficient decreases. The results can play role in elemental material analysis, especially in archaeometry.

## 1. Introduction

The Kβ/Kα intensity ratios have been intensively studied since 1969. Daoudi et al. [1] reports that after more than thousand measurements, 127 theoretical and experimental publications have been created in the last half century. The experimental measurements of X-ray spectra are crucial in examining theoretical models [2,3,4]. The values of Kβ/Kα intensity ratios are used for estimation the vacancy transfer probability (e.g., transfer of hole from *K* shell to *L* shell [5,6,7])) and Kα and Kβ X-ray production cross-sections [8,9,10,11]. In experiments the excitation mediums were mainly radioisotopes: ^241^Am [10,12,13,14,15,16,17,18,19,20,21,22,23,24,25,26,27,28], ^109^Cd [29,30,31], ^137^Cs [32,33], ^57^Co [34], and ^238^Pu [35]. X-ray tubes were also used for Kβ/Kα intensity ratios studies [36,37,38,39] as well as proton beam [40,41].

There are a couple of aspects linked to the study on Kβ/Kα intensity ratios. At first, experiments in which chemical compounds were tested showed that the Kβ/Kα X-ray intensity ratios are sensitive to the chemical environment for 3d elements [13,20,42,43]. The results were explained by the change in screening of 3p electrons by 3d valence electrons. The studies on 3d metal alloys and compounds have shown dependence of the Kβ/Kα intensity ratios on alloy composition or chemical state through changes in electron binding and electron configuration of the valence states [16]. Since Cu 3d states do not overlap energetically with the Ag 4d band, energy mismatch between Ag 4d and Cu 3d states is the main contributor to the sharpness and degeneracy of the Cu 3d states. Despite the lack of overlap of silver and copper wave functions in the Ag–Cu alloy, the charge density is transferred between Ag and Cu [44]. Raj et al. [17] report that in alloys the 3d electron transfer/delocalization is the main factor causing change in the Kβ/Kα intensity ratios. Thus the changes in Kβ/Kα intensity ratios of alloy’s element indicate changes in the valence electronic configurations or charge transfer effect caused by presence of second elements [25]. The measurement of Kβ/Kα intensity ratios can be a sensitive probe of 3d charge transfer [45].

Another advantage of Kβ/Kα X-ray intensity ratio studies is that this parameter can be used for determination of depth profile distributions of the elements in thick targets [40]. This technique can be used in archaeometry to determine the silver enrichment taking place in antique silver–copper coins [39,46,47,48]. Since many decorative objects are composed of silvered copper, gilded copper, or silver, the use of Kβ/Kα X-ray intensity ratio from different chemical element can allow estimating the thickness of the surface layer [49].

The knowledge on Kβ/Kα X-ray intensity ratio can also be a tool to find elements which *K* or *L* lines overlap the lines of main element. So the Kβ/Kα X-ray intensity ratio can be used to find the intensities of the unresolved lines of neighbor elements [50].

Research on Kβ/Kα intensity ratios of complex materials motivated us the use of advanced Monte-Carlo tools, by means of the FLUKA code [51,52], to simulate complex X-ray spectra. The Monte-Carlo simulation method was introduced in 1949 and since then it has been successively used in many areas of physics, such as atomic physics, high energy physics, medical physics, as well as in material engineering, construction of accelerator structures, and in other fields of science, including mathematics, biology, economics, and archaeometry. Although the FLUKA code does not include in-depth quantum-mechanic features at atomic level such as the charge transfer between 3d electrons, in our paper the analytical challenge is based on a comprehensive and accurate description of the spectrum features such as the shape of the primary radiation spectrum, i.e., the intensity and the shape of the X-ray tube anode lines, and the intensity, the centroid, and the shape of all emission lines of the tested material. Each of the individual elements of the spectrum provides relevant analytical information. The simulation allows the determination of the Kβ/Kα intensity ratios without the need to transform the radiation intensity of the characteristic sample obtained in the detector. Self-absorption and detector performance corrections, which are usually necessary in conventional quantitative analysis based on main peak analysis, are therefore eliminated, which means that the FLUKA code just simulates an experimental output, not a detector input. The reliability of Monte-Carlo tools, in addition to the subjective modeling of the composition and structure of the sample, depends on the analytical model adopted, the description of the radiation source, and the settings of equipment specifications, operational parameters, and experimental geometry. Since Kβ/Kα intensity ratios of Cu have been extensively explored the simulation of this element is an appropriate test point.

In this work, X-ray spectra of silver–copper alloys were modeled. The copper Kβ/Kα intensity ratios were calculated for pure Cu as well as for Ag–Cu alloys. Two kinds of the FLUKA simulations have been performed. The first kind includes primary electron beam and radiation of X-ray tube equipped with a Rh anode operating at 40.8 kV. The second kind includes a monoenergetic 59.9 keV photon beam. The fluorescence spectra of silver–copper alloys are an output for both kind of simulations. The results obtained are critically evaluated by comparison with available experimental and theoretical values for pure elements. It is worth underlining that the obtained results are not exactly the same kind as the experimental results. It is because the experimental results are based on the X-ray photons counted by the detector and then the values are corrected by detector efficiency and air and sample absorption coefficients. In contrast, the FLUKA simulation results are based on the X-ray photons emitted directly from the sample.

## 2. Experiment Simulations

Monte-Carlo simulations were performed using FLUKA 2011 code version 2c.8 installed on a computer cluster at Świerk Computing Center [53]. FLUKA code uses the Evaluated Photon Data Library (EPDL97) [54]. The EPDL library consists of tabulations of photon interaction data including photoionization, photoexcitation, coherent and incoherent scattering, and pair and triplet production cross sections.

The experimental setup reproduced in the calculations, consisting of an X-ray tube model with a 1 mm thick Rh anode, a 1 mm thick Be window, and two irradiated sample groups with a diameter of 1 cm and a thickness of 2 mm and 1 µm, is shown in Figure 1. Additionally, for samples with thickness of 1 µm the monoenergetic ^241^Am (59.9 keV) have been used. As one can see from Figure 1, in the case of 1 µm sample a part of radiation is going through the sample and another part is reflected back off the sample. The first part is called the forward output flux and the second part is called the backward output flux (this part is usually incoming to the detector). In the case of the 2 mm sample there is no forward output flux, because all radiation going trough the sample is absorbed or re-emitted in a backward direction. Calculations for each alloy were made by dividing them into 500 parallel processes. The 1 keV photon and electron transport energy cut-off was set to best reproduce photon and electron behaviour for the used beam energy range. The Rh-X-ray was induced by a $2\cdot 10^{11}$ monoenergetic electron beam (40.8 keV) with flat distribution, Φ = 1 cm. The Rh anode X-ray spectrum filtered by a 1 mm Be layer is presented in Figure 2. The calculated K X-ray spectra of Ag–Cu alloy registered on a flat, irradiated sample surface are presented in Figure 3.

## 3. Results and Discussion

The simulated Cu Kα and Cu Kβ intensities as well as the different Ag–Cu alloys are presented in Table 1, Table 2 and Table 3. In our work we have considered the following K-x-ray transitions: $K\alpha_{1,2}$ (*K*-$L_{2,3}$), $K\beta_{1,3}$ (*K*-$M_{2,3}$), and $K\beta_{2}$ (*K*-$N_{2,3}$). The dependence of Kβ/Kα intensity ratio on copper concentration in Ag–Cu alloy is presented in Figure 4, Figure 5 and Figure 6. Three cases are studied: backward output flux, forward output flux, and weighted average output flux. The error bars arise from statistical uncertainties. In real experiment the errors are attributed to uncertainties from various parameters used in the determination of the Kβ/Kα intensity ratio, including errors caused by the evaluation of peak area, detector efficiency, self-absorption factors, target thickness, and counting statistic. Table 4 presents the available theoretical and experimental values for pure copper while in Table 5 values for Cu alloys obtained from available literature. The literature data for pure Cu and Cu–Ag alloy are also presented collectively in Figure 4, Figure 5 and Figure 6.

Some general conclusions can be drawn based on the presented data: (i) In the case of backward and average output fluxes, there is very small difference between results calculated for 1 µm sample for both Rh X-ray tube and monoenergetic 60 keV radiations, but here is distinct difference between these results and results calculated for the 2 mm sample. (ii) There is no forward output flux in the case of the 2 mm thick sample, because all radiation is absorbed in this direction while moving through the sample. For 1 µm samples the difference between results for Rh X-ray tube and monoenergetic radiations is bigger than in the case of backward output flux. (iii) As can be seen from Figure 4, a major part of experimental results is placed in between the FLUKA results calculated for very thin (1 µm) and very thick (2 mm) samples. The present results can also partially explain the differences between various experimental results for pure copper as a result of different thickness of samples used in experiments. (iv) The Cu Kβ/Kα intensity ratio is sensitive to alloy composition. As the Cu content increases, the Kβ/Kα coefficients decrease. The alloying effect is in order of a few percent and this size of effect is consistent with the size of the alloying effect reported by Dhal et al. [28].

## 4. Conclusions

The Cu Kβ/Kα intensity ratios for pure copper and for a sequence of nine Ag–Cu alloys (from 10% to 100% Cu) have been simulated with the FLUKA code. The results can play role in elemental material analysis, especially in archaeometry. Silver and copper are used in jewelry and minting from antique times [39,58,59,60]. Thus it is in the interest of archaeologists to explore the ancient technologies of silver jewelry production. Copper was often added to silver to make sterling silver, increasing its strength. The concentration of more than 2.6% Cu indicates a deliberate addition by ancient manufacturers. The spectroscopic techniques like ED-XRF, SEM-EDX, or PIXE are commonly used in compositional research. The elemental content is determined by using intensity of peaks recorded in energetic spectra. However, these techniques can be used for surface and subsurface analysis. The Kβ/Kα X-ray intensity ratio analyses can be applied for elemental composition analysis as well as for determination of depth profile distributions of the elements in studied artifacts. The thickness of coating in double layers artifacts and silver surface enrichment of silver–copper alloys can be also determined. Moreover, since the Ag–Cu alloying system has many other applications, among others it is often used in nanotechnology [61,62] and it is estimated as the best material for improving oxidation resistance with only a slight reduction in electrical conductivity [63], knowledge about alloying effects may play important role in those areas.

## Figures and Tables

**Figure 1 materials-14-04462-f001:**
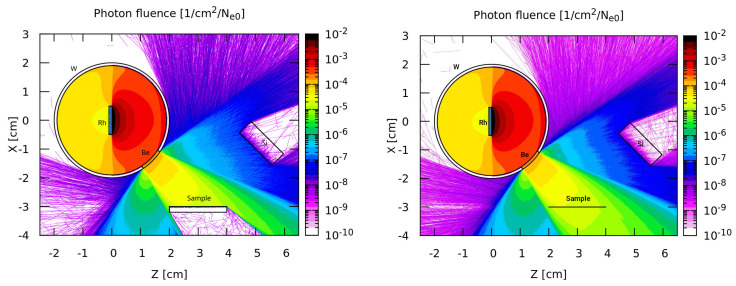
Experimental setup and photon fluence reproduced in the calculations for the sample thickness of 2 cm (**left**) and 1 µm (**right**).

**Figure 2 materials-14-04462-f002:**
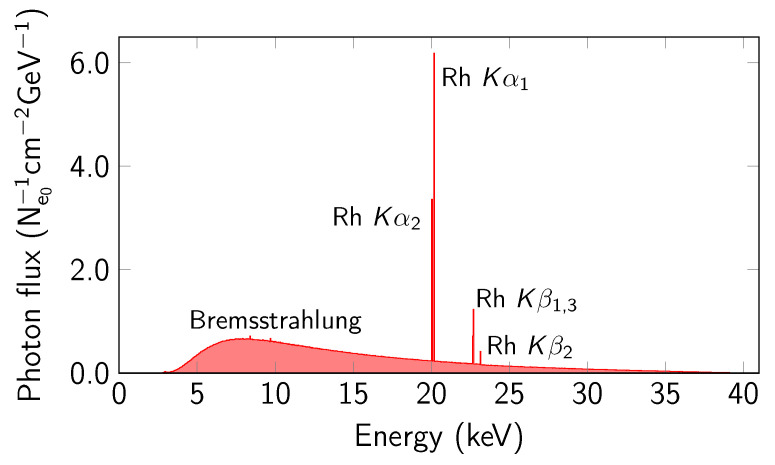
Rh anode X-ray spectrum.

**Figure 3 materials-14-04462-f003:**
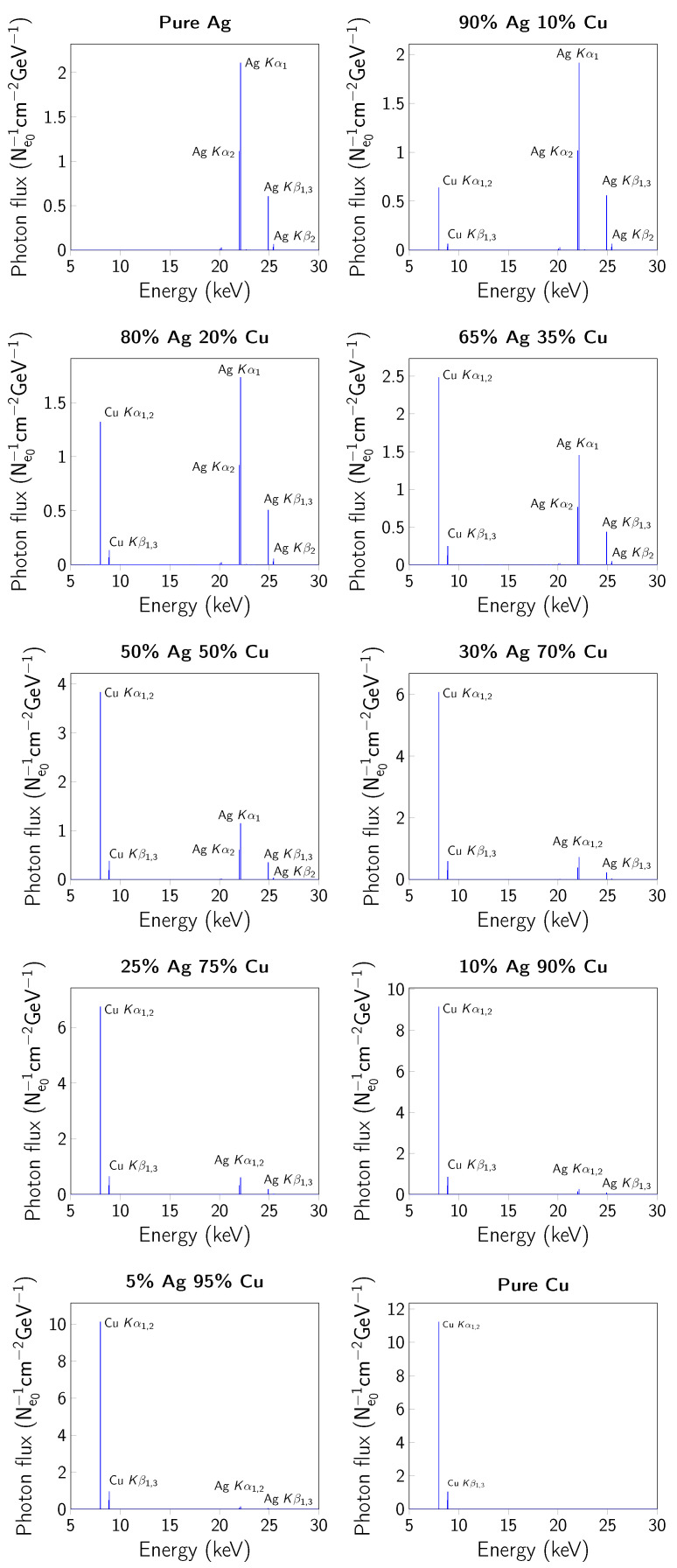
X-ray spectra in studied Ag–Cu alloys.

**Figure 4 materials-14-04462-f004:**
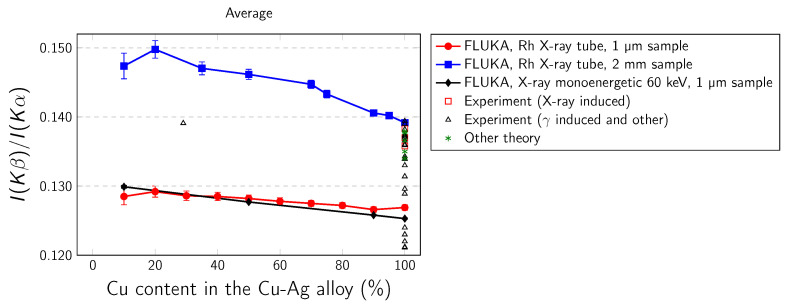
Kβ/Kα intensity ratio arising from the primary X-ray simulated with FLUKA, calculated for average output flux in 1 µm and 2 cm samples.

**Figure 5 materials-14-04462-f005:**
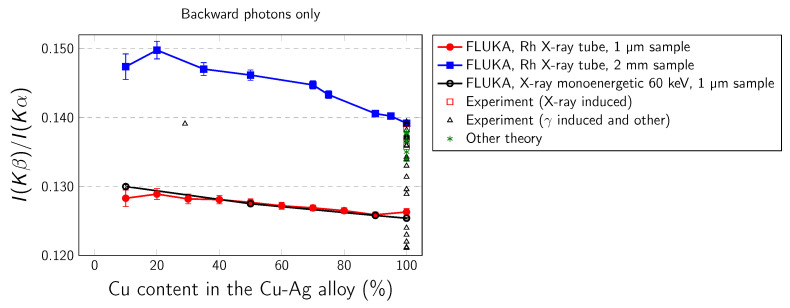
Kβ/Kα intensity ratio arising from the primary X-ray simulated with FLUKA, calculated for backward output flux in 1 µm and 2 cm samples.

**Figure 6 materials-14-04462-f006:**
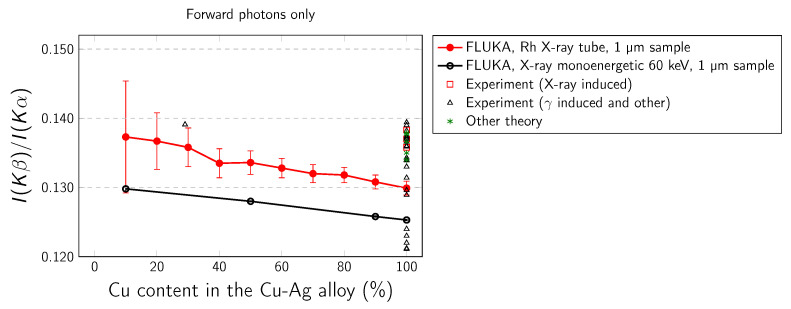
Kβ/Kα intensity ratio arising from the primary X-ray simulated with FLUKA, calculated for forward output flux in 1 µm sample.

**Table 1 materials-14-04462-t001:** Simulated Kβ/Kα intensity ratio for 1 µm thick samples induced by Rh X-ray tube radiation, calculated for backward and forward direction and weighted average of them.

Cu (%)	Kβ/Kα		
	Average	Backward Only	Forward Only
10	0.1285(12)	0.1283(12)	0.1373(81)
20	0.1292(8)	0.1289(8)	0.1367(41)
30	0.1286(7)	0.1282(7)	0.1358(28)
40	0.1285(6)	0.1281(6)	0.1335(21)
50	0.1282(5)	0.1277(5)	0.1336(17)
60	0.1278(5)	0.1272(5)	0.1328(14)
70	0.1275(4)	0.1269(4)	0.1320(13)
80	0.1272(4)	0.1265(4)	0.1318(11)
90	0.1266(4)	0.1259(4)	0.1308(10)
100	0.1269(4)	0.1263(5)	0.1299(10)

**Table 2 materials-14-04462-t002:** Simulated Kβ/Kα intensity ratio for 1 µm thick samples induced by monoenergetic radiation, calculated for backward and forward direction and weighted average of them.

Cu (%)	Kβ/Kα		
	Average	Backward Only	Forward Only
10	0.1299(2)	0.1300(3)	0.1298(3)
50	0.1277(1)	0.1275(1)	0.1280(1)
90	0.1258(1)	0.1258(1)	0.1258(1)
100	0.1253(1)	0.1254(1)	0.1253(1)

**Table 3 materials-14-04462-t003:** Simulated Kβ/Kα intensity ratio for 2 mm thick samples induced by Rh X-ray tube radiation. Only backward direction is calculated.

Cu (%)	Kβ/Kα
10	0.1474(19)
20	0.1498(13)
35	0.1470(9)
50	0.1462(7)
70	0.1447(6)
75	0.1433(6)
90	0.1406(5)
95	0.1402(4)
100	0.1392(4)

**Table 4 materials-14-04462-t004:** Kβ/Kα intensity ratio for copper taken from the literature.

Kβ/Kα	Reference	Excitation Source
Experiment:		
0.1382(16)	[12]	^241^Am
0.1370(110)	[13]	^241^Am
0.1330(33)	[14]	^241^Am
0.1212(90)	[10]	^241^Am
0.1211(19)	[15]	^241^Am
0.1360(6)	[17]	^241^Am
0.1340(130)	[18]	^241^Am
0.1343(12)	[19]	^241^Am
0.1374(113)	[20]	^241^Am
0.1390(130)	[21]	^241^Am
0.1220(100)	[22]	^241^Am
0.1360(60)	[13]	^241^Am
0.1359(30)	[24]	^241^Am
0.1314(87)	[25]	^241^Am
0.1289(86)	[26]	^241^Am
0.1296(66)	[27]	^241^Am
0.1360(10)	[28]	^241^Am
0.1394(70)	[55]	^241^Am
0.1360(60)	[35]	^238^Pu
0.1366(330)	[56]	^109^Cd
0.1370	[30]	^109^Cd
0.1390(56)	[31]	^109^Cd
0.1240(30)	[33]	^137^Cs
0.1240(90)	[32]	^137^Cs
0.1339	[34]	^57^Co
0.1360(20)	[38]	*K*-capture
0.1372(10)	[41]	1 MeV protons
0.1358(17)	[36]	50 kV W X-ray tube
0.1383(55)	[37]	35 mA W X-ray tube
0.1370(20)	[38]	30 kV Mo X-ray tube
0.123(7)	[57]	10 keV synchrotron radiation
Theory:		
0.1379	[2]	
0.1377	[3]	
0.1340, 0.1350, 0.1366, 0.1377	[4] *	

* different approaches.

**Table 5 materials-14-04462-t005:** Kβ/Kα intensity ratio and relative Kβ/Kα ratio (compared to Kβ/Kα ratio of pure Cu) for copper alloys taken from literature.

Cu Alloy	Kβ/Kα	Relative Kβ/Kα	Reference
Cu_29_Ag_71_	0.1391(7)	-	[28]
Cu_94_Sn_6_	0.1351(6)	-	[28]
Cu_48.4_Sn_51.6_	0.1419(72)	1.0949	[27]
Cu_14_Sn_86_	0.1429(73)	1.1026	[27]
Cu_6.1_Sn_93.9_	0.1381(70)	1.0656	[27]
Co_25_Cu_74_Ag_1_	0.1388(92)	1.0563	[25]
Co_31_Cu_68_Ag_1_	0.1444(96)	1.0989	[25]
Co_36_Cu_63.6_Ag_0.4_	0.1371(91)	1.0434	[25]
Co_10.7_Cu_89.1_Ag_0.2_	0.1341(89)	1.0205	[25]
CuAl	0.1335(6)	-	[16]

## Data Availability

The data presented in this study are available on request from the corresponding author.

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
