# Peer review of "FLUKA Simulations of *Kβ*/*Kα* Intensity Ratios of Copper in Ag–Cu Alloys"

_materials, 2021, doi:10.3390/ma14164462_

Round 1
Reviewer 1 Report
The paper reports on the determination of the K-beta/K-alpha ratio in copper and copper-silver alloys. They quote a number of application that rely on this ratio, notably the detection of surface enrichment of silver alloys, which justifies relevance of their work. Further, one should add that this ratio is extremely important in X-ray fitting programs that usually fit aggregates of selected X-ray lines. The authors found a reasonable variance of the K-beta/K-alpha ratio in copper-silver alloys, pointing an increase of about 5% in Cu(10%) Ag(90%) alloy. They interpret this by changes in the configuration of valence electrons, which involves charge transfer between 3d subshell electrons in both elements.
Though this effect has been detected experimentally, an extreme caution should be paid to this statement. As the present results were not obtained by experiment, but by Monte Carlo simulation using the FLUKE program, the authors shall carefully check the program documentation if the codes indeed take into account the charge transfer between 3d electrons. If not, the dependence of K-beta/K-alpha ratio as a function of Cu content may rather be result of incomplete absorption correction in thick targets. The correction term for monoenergetic excitation is, for example, given by eqs. (2,3 ) in Dhal et al. 1994 (ref. 28; note that in their experiment, the sample was directly facing the X-ray detector. If not so, additional cos of the X-ray take-off angle has to be added. In Fig. 1 of the simulation experiment, the impact and take off angles seem the same, 45 degrees, so the co-sinuses cancel in eq. 3). For excitation with the X-ray tube, the correction term has to be integrated over photon energy distribution. However, a simpler approach (for this review) may just rely non some mean impact energy. Two examples are shown in the attached figure, 9 keV just above the absorption edge, and 24 keV, which was selected subjectively in the middle of the energy distribution above Cu absorption edge. The ratio between the correction factors for the alloy composition and for pure copper then reasonably well reproduces the values, given in the second column of Table 1. As a corroboration, the authors’ values are in reasonable disagreement with the value of Dhal et al. for Cu(29%) Ag(71%) alloy: their K-beta/K-alpha ratio is essentially the same as in pure copper.
The authors further evaluate the vacancy transfer probability according to the simple relation of Schonfeld (ref. 56). It is questionable if all digits have sense, as the fluorescence yield may also change due to alloying effect (ref. 25).
Though the paper is well written in understandable English and shows a rich literature survey, the authors shall revisit the charge transfer explanation and absorption correction. Accordingly, they would probably reformulate the sentences 53-56.
Some smaller linguistic remarks:
line 12: from 1969 – since 1969 (stylistic)
line 16: estimation the – estimation of the
line 26: band energy mismatch – band, the energy mismatch
line 107: by use – by using

Author Response
Reply to the Report of the Referee 1
We would like to thank the Referee for reviewing the manuscript and positive feedback.
Below we have answered the Referee’s questions:
Referee’s comment:
The authors found a reasonable variance of the K-beta/K-alpha ratio in copper-silver alloys, pointing an increase of about 5% in Cu(10%) Ag(90%) alloy. They interpret this by changes in the configuration of valence electrons, which involves charge transfer between 3d subshell electrons in both elements. (…) As the present results were not obtained by experiment, but by Monte Carlo simulation using the FLUKE program, the authors shall carefully check the program documentation if the codes indeed take into account the charge transfer between 3d electrons.
Answer:
The FLUKA code does not incorporate the effect of the charge transfer between 3d electrons. Although, as stated in the Introduction, the change of K-beta/K-alpha ratio is often explained by 3d electron transfer, we had a look on the other points such as self-absorption coefficient change while alloy composition changes and source geometry setup.
Referee’s comment:
(…) absorption correction in thick targets (...)
Answer:
Indeed, the FLUKA simulations are based on absorption corrections of materials. To study this aspect deeper, we perform more simulations as Referee 2 suggested.
Referee’s comment:
As a corroboration, the authors’ values are in reasonable disagreement with the value of Dhal et al. for Cu(29%) Ag(71%) alloy: their K-beta/K-alpha ratio is essentially the same as in pure copper.
Answer:
The disagreement is probably a result of sample thickness. As one can see from a new Fig. 4, the experimental result of Dhal et al. is in the middle of the results of simulations performed for thick (2 mm) and thin (1 µm) target sample. In fact, most of experimental results are in the middle of our results.
Referee’s comment:
The authors further evaluate the vacancy transfer probability according to the simple relation of Schonfeld (ref. 56). It is questionable if all digits have sense, as the fluorescence yield may also change due to alloying effect (ref. 25).
Answer:
We removed the numbers for vacancy transfer probability from the manuscript.
Referee’s comment:
the authors shall revisit the charge transfer explanation and absorption correction. Accordingly, they would probably reformulate the sentences 53-56.
Answer:
We improved slighty in the Introduction section and removed misleading sentence from the Abstract.

Reviewer 2 Report
Comments in attachment.

Author Response
Reply to the Report of the Referee 2
We would like to thank the Referee for reviewing the manuscript and positive feedback.
Below we have answered the Referee’s questions:
Referee’s comment:
There must be information explaining the source of the X-ray parameters used in FLUKA. (...) The manuscript must answer if, for exemple, FLUKA uses xray lib?
Answer:
FLUKA does not use xraylib library. For atomic data such as tabulations of photon interaction data including, photoionization, photoexcitation, coherent and incoherent scattering, and pair and triplet production cross sections FLUKA uses EDPL97 library.
Referee’s comment:
The introduction contains an interesting discussion of the 3d annealing, however it should be already evident in the introduction if FLUKA incorporate these effects or not.
Answer:
The FLUKA code does not incorporate the effect of the charge transfer between 3d electrons. We agree that the sentences in the Introduction about 3d charge transfer may mislead about the real work did in the manuscript. We improved slighty in the Introduction section and removed misleading sentence from the Abstract.
Referee’s comment:
It is not clear if the obtained ratios are only caused the x-ray parameters used in FLUKA, or if there are geometrical effects or the Rh x-ray tube profile playing a role. The authors must repeat the simulations but for a simple setting only dependent on the x-ray parameters, such as a monoenergetic and unidirectional beam of x-rays (like241Am), and for at least three proportions of Cu and Ag.
Answer:
Inspired by reviewer we made new simulations for thin 1 µm Ag-Cu alloy. As a irradiation source we use Rh-X-Ray tube as well as 241Am and we obtained comparable results, but we found discrepancy between previous results for thick (2 mm) sample and a new thin sample. The results of these simulations have been added to the manuscript. Due of limited time assigned to the Reply to the Referee we performed only limited calculations. In the near future we would like to continue the simulations for different thickness to examine the thickness effect in more details.
Referee’s comment:
Finally there are no comments or explanations for the differences between the obtained ratios and the experimental ones. Is this due to geometrical effects or the x-ray parameters of FLUKA?
Answer:
The disagreement is probably a result of sample thickness. As can be seen from a new Fig. 4, a major part of experimental results is placed in between the FLUKA results calculated for very thin (1 µm) and very thick (2 mm) samples. Then we guess that the different thickness of samples used in experiments is the main, or at least a one of the main reasons, of difference between FLUKA simulations and experiments and between various experiments.
Referee’s comment:
Figure 4 should contain the experimental points of table 2.6. In the abstract, the mention of a strong correlation of the Cu to Ag content to the ratio is wrong. 5% relative difference due Cu to Ag content is not strong.
Answer:
We added experimental points to Fig. 4. About the word “strong” – we changed it to the word “visible”.

Round 2
Reviewer 1 Report
The authors have largely improved the manuscript with the new calculations for thin targets. The differences in K-beta/K-alpha ratios in thin and thick targets are now correctly interpreted as consequence of X-ray absorption, and not of the fineness of orbital interactions.
However, the influence of the absorption is still not well explained. The results of the simulation are the K-beta/K-alpha ratios that an actual experiment will produce, safe for using an ideal X-ray detector in vacuum. This means that the obtained ratios cannot be compared to the values of an isolated atom, obtained by atomic calculations or deduced from the experiment carefully taking into account all thick target effects. I propose the authors clearly state that the obtained results are simulation of the experimental values, and that Monte Carlo simulation was necessary on account of the complex excitation spectrum and experimental geometry. After that, they may further apply an analytical thick target correction factor (like the one mentioned and calculated in my previous report) to remove the influence of absorption, and only then compare the values to the literature data. Some absorption is present also for the thinnest targets, as demonstrated by the slope of the curves.
Other comments: The authors shall mark in Fig.1 how they detected forward and backwards radiation. In lines 93, 94, it is a bit misleading to compare backward and average fluxes: their agreement is consequence of big statistical errors of the forward flux, which actually did not enter calculation.
Some other linguistic errors and suggestions:
line 23: band energy mismatch -> band, the energy mismatch
line 28: valance -> valence
lines 34, 35: .. silver the using of -> silver, the use of (or silver, using of)
Figure 1: … reproduced in the calculations for the sample thickness of 2 cm (left) and 1 μm (right)
lines 51-54: without the need to transform the radiation intensity … Self-absorption and detector … are therefore eliminated. - What does this mean? That an ideal detector was assumed and target absorption correction was skipped, just to simulate an experimental output?
Figure 6: add ‘in 1 μm sample.’
line 103: depends visible -> is sensitive
line 132: deliberate action -> deliberate addition by ancient manufacturers.
line 134: by use -> by using
Author Response
I would like to thank the Refereesfor reviewing the manuscript again.
We improved our manuscript following the report of Referee. We fixed all mentioned linguistic errors. About "forward and backwards radiation" we add appropriate sentences in Section 2 instead of modify the Figure 1. About "state that the obtained results are simulation of the experimental values" and "compare the values to the literature data" - we added appropriate sentences at the end of Section 1. Unfortunately, we are not able to compare our results to all particular experiments cited taking into account all correction factors, without knowing the thickness of samples used in the experiments. Unfortunately, not all experimental papers provide all information needed. Instead of comparing our results to the particular experiments, we show how the difference between experimental results could be explained by the difference of samples thickness.
Reviewer 2 Report
The authors took into account my criticism and I approve the manuscript in the present form.
Author Response
I would like to thank the Referee for reviewing the manuscript again.